# Imputation of Missing Behavioral Measures in Connectome-based Predictive Modelling

**Qinghao Liang** [1]   **Dustin Scheinost** [1,2]

## Abstract

The best performing, connectome-based predictive models of behavior use multiple sources of data (*e.g.,* predicting latent variables generated from a battery of behavioral measures). However, as the number of sources increases, the chances of missing a portion of the behavioral measures also increases, hindering downstream analyses. The most common strategy for handling missing data is to remove participants with missing values and run the analysis only using the complete cases. This approach hinders downstream predictive modeling algorithms that rely on large data sets for training. To allow participants with missing data to be retained for training, we included a data imputation step in connectome-based predictive modeling (CPM) to estimate missing values in the behavioral measures. Performance is evaluated by the improvement of predicting power compared with complete case study. Experimental results show that imputation of missing behavioral measures improves CPM performance when the predictability of that behavioral measure is relatively high. Overall, our results suggest that increasing the size of training data via data imputation may be a valuable step for datasets with missing behavioral data.

## 1. Introduction

Advanced functional magnetic resonance imaging (fMRI) techniques—especially, functional connectivity matrices, or connectomes—are revealing robust individual differences in behavior (Dubois & Adolphs, 2016). Further, emerging works are beginning to highlight the benefits using multiple sources of information per participant in detecting these individual differences (Gao et al., 2019a; Dadashkarimi et al., 2019; Elliott et al., 2019; Dubois et al., 2018). For example, models combining multiple connectomes outperform models built from a single connectome (Gao et al., 2019a) and latent variables derived from a battery of behavioral measures are more predictable than a single behavioral measure (Dubois et al., 2018; Gao et al., 2019b). Together, these results suggest that using connectomes and behavioral data from multiple sources is a powerful approach to modeling individual differences. Nevertheless, using multiple sources of data increases the likelihood of missing data. Currently, most fMRI studies only consider complete cases. In other words, participants with missing behavioral or imaging data are simply removed from analysis, introducing potential selection biases, reducing statistical power, and hurting generalization.

In this work, we introduce a data imputation step to connectome-based predictive modeling (CPM) to improve brain-based models of behavior by including participants with missing data in model training. While previous works have explored data imputation for missing imaging data (Vaden et al., 2012; Thung et al., 2018; Yuan et al., 2012), using imputation for missing behavioral data is less explored. Specifically, we evaluate three data imputation methods (mean imputation, regularised iterative principal component analysis (Josse & Husson, 2016), and missForest (Stekhoven & Bühlmann, 2011) to impute either a single behavioral measure or a latent behavioral factor from a battery of behavioral measures. Participants with imputed behaviors are then retained to train brain-based models of behavior using 10-fold cross-validation and ridge regression CPM (Gao et al., 2019a). As the goal of the imputation is to retain more participants for building downstream CPM models, we assess the utility of each imputation method as the change in CPM performance compared to a case where all participants with missing data are removed (*i.e.,* the complete case study). To do so, we simulate missing data (completely at random) in two large open-source datasets: the Human Connectome Project (HCP) dataset (Essen et al., 2013) and the UCLA Consortium for Neuropsychiatric Phenomics (CNP) dataset (Poldrack et al., 2016)

[1] Department of Biomedical Engineering, Yale University, New Haven, CT, USA [2] Department of Radiology and Biomedical Imaging, Yale School of Medicine, New Haven, CT, USA. Correspondence to: Dustin Scheinost <dustin.scheinost@yale.edu>.

*Presented at the first Workshop on the Art of Learning with Missing Values (Artemiss) hosted by the $37^{th}$ International Conference on Machine Learning (ICML).* Copyright 2020 by the author(s).

# 2. Method

## 2.1. Ridge regression Connectome-based Predictive Modeling (rCPM)

CPM is a validated method for extracting and pooling the most relevant features from connectivity data in order to construct linear models to predict phenotype measures (Shen et al., 2017). Each connectome is vectorized and the edges are taken as features. Then edges of connectivity matrices that are significantly correlated with the phenotypic measure of interest are selected. In ridge regression CPM, a ridge regression model is directly fitted with training individuals using the selected edges from multiple task connectomes per individual and the model is applied to testing individuals in the cross-validation framework. Due to the positive semi-definite nature of a functional connectivity matrix, the edges are not independent. Ridge regression is more robust than ordinary least-squares regression when dealing with dependent features (Gao et al., 2019a).

## 2.2. Data imputation methods

To address the missing data problem in behavioral measures and incorporate more subjects into training, we tested three different data imputation methods in the setting of predicting one single behavioral measure and predicting the latent phenotype of multiple behavioral measures using rCPM. In most functional connectivity studies, there are multiple behavioral measures $Y = \{y_1, y_2, ..., y_n\}$, where each $y_i$ is a column vector. Thus, it is possible to impute the missing values using some developed imputation method simply on the behavioral data.

### 2.2.1. MEAN IMPUTATION

In mean imputation, missing values were replaced with the mean of the non-missing entries of that variable.

### 2.2.2. MISSFOREST

MissForest is a non-parametric missing value imputation method for mixed-type data. For each variable in the dataset, missForest fitted a random forest on the observed part and then predicted the missing part (the predicted values were later used in training models of other variables). The algorithm continued to repeat these two steps until a stopping criterion was met. The algorithm was implemented in R package **missForest** (Stekhoven & Bühlmann, 2011).

### 2.2.3. REGULARIZED ITERATIVE PRINCIPAL COMPONENT ANALYSIS

Iterative Principal Component Analysis (PCA) algorithm, also known as the Expectation-maximization PCA (EM-PCA) algorithm, is an expectation-maximization algorithm

for a PCA fixed-effects model, where data are generated as a fixed structure having a low rank representation corrupted by noise. Regularized iterative Principal Component Analysis used regularized methods to tackle the overfitting problems when data is noisy and/or there are many missing values (Verbanck et al., 2015). The algorithm was implemented in the function **imputePCA** in R package **missMDA** (Josse & Husson, 2016).

## 2.3. Simulation

To test if the imputation methods improved rCPM performance, we simulated the missing data completely at random behavioral measures. The missing percentage was varied from 2.5% to 40% in increments of 2.5%. At each missing percentage, we performed 100 trials using different seeds to generate synthetic missing data. In all cases, rCPM with 10-fold cross validation and 10 repeated random splits was used to generate predicted values. Additionally, the three imputation methods described above were used to impute missing data using only the training data for each fold of cross validation.

### 2.3.1. SINGLE BEHAVIORAL MEASURE

Here, we aimed to predict $y_k$, which has some missing elements. The missing percentage of the target variable $y_k$ and the rest of data were controlled separately with the same value. Next, we separated the dataset into two splits $\{X, Y\}_{miss}$ and $\{X, Y\}_{obs}$. $\{X, Y\}_{miss}$ contains the data of participants whose $k^{th}$ behavioral measure is missing. In contrast, $\{X, Y\}_{obs}$ contained the data of participants whose $k^{th}$ behavioral measure was complete. In the 10-fold cross-validation of rCPM, $\{X, Y\}_{obs}$ was divided into 10 groups. On each fold, 9 of these folds were combined with $\{X, Y\}_{miss}$ as training set $\{X, Y\}_{train}$. The missing values of $y_k$ were then imputed in $\{Y\}_{train}$. Then the model is trained on now completed $\{X, y_k\}_{train}$ and validated on $\{X, y_k\}_{test}$.

### 2.3.2. LATENT PHENOTYPE

For predicting the latent phenotype (the $1^{st}$ principal component), the average missing percentage of all behavioral measures was controlled. The whole dataset was directly divided into $\{X, Y\}_{train}$ and $\{X, Y\}_{test}$ in each fold. $\{Y\}_{train}$ and $\{Y\}_{test}$ are imputed separately. After filling in the missing data, we applied principal component analysis (PCA) on $\{Y\}_{train}$ to get the first principal component $y_{pc_{train}}$. The PCA coefficients were subsequently applied to $\{Y\}_{test}$ to get $y_{pc_{test}}$.

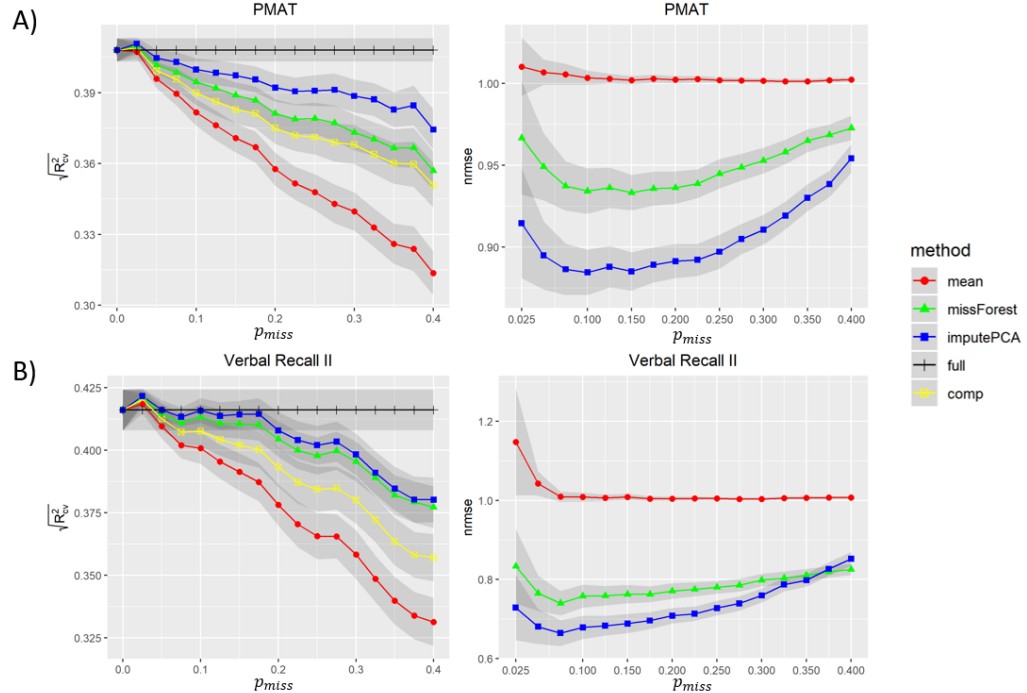

*Figure 1.* Performance of rCPM with embedded data imputation in predicting A) PMAT (HCP dataset) B) Verbal Recall II (CNP dataset). Prediction performance ($\sqrt{R_{CV}^2}$) and imputation accuracy ($nrmse$) over a range of missing data rates from 2.5% to 40% missing data are shown. The shadow areas represent the 95% confidence interval calculated from multiple repeats of missing data.

### 2.4. Comparison methods

Prediction performance was evaluated by the cross-validated $R^2$, $R_{CV}^2 = 1 - \frac{\sum_{i=1}^{n}(y_i - \hat{y})^2}{\sum_{i=1}^{n}(y_i - \bar{y})^2}$ (Alexander et al., 2015). $\sqrt{R_{CV}^2}$ was reported for comparability to the normally-used Pearson correlation coefficient. Normalized mean squared error was used to compare the imputed values to the ground truth, $nrmse = \sqrt{\frac{mean((y_k^{true} - y_k^{imp})^2)}{var(y_k^{true})}}$, where $y_k^{imp}$ is the imputed value for missing entries and $y_k^{true}$ is the true value. 95% confidence intervals were calculated as: $\bar{x} \pm t^* \frac{s}{\sqrt{n}}$, where $t^*$ is the upper $0.025$ critical value for the $t$ distribution with $n - 1$ degrees of freedom, $n$ is the sample size, and $s$ is the estimated standard deviation. All results using data imputation were compared to the complete case results, where any individual with missing data is simply dropped from analysis.

## 3. Result

### 3.1. Dataset

#### 3.1.1. HCP DATASET

In this dataset, each individual performed seven tasks in the scanner. The seven task scans (gambling, language, motor, relational, social, working memory, and emotion)

were processed with standard methods and parcellated into 268 nodes using a whole-brain, functional atlas, as previously described (Glasser et al., 2013; Gao et al., 2019a). Functional connectivity was calculated based on the "raw" task timecourses, with no regression of task-evoked activity. Next, the mean timecourses of each node pair were correlated and correlation coefficients were Fisher transformed, generating seven $268 \times 268$ connectivity matrices per subject. Cognitive ability was assessed by tasks from the NIH tool-box and Penn computerized neurocognitive battery. After excluding participants for high motion or incomplete data, 500 subjects were retained for simulations.

For the single behavioral variable experiments, we chose variables, which could be predicted with a high prediction accuracy (Penn Matrix Reasoning Test (PMAT), ReadEng and PicVocab). For the latent factor experiments, we used the unadjusted score of ten behavioral measures from Dubois *et al.* (Dubois et al., 2018) (PicVocab, PMAT, ReadEng, VSPLOT, IWRD, PicSeq, ListSort, Flanker, CardSort and ProcSpeed). All behavioral measures selected are correlated with intelligence.

#### 3.1.2. CNP DATASET

In this dataset, each individual performed six tasks in the scanner. The six task scans (balloon analog risk task, paired

associative memory encoding, paired associative memory retrieval, spatial working memory capacity, stop signal, and task switching) were processed in the same way as above. After excluding participants for high motion or incomplete data, 172 subjects were retained for simulations.

Similar to the HCP dataset, for the single behavioral variable experiments, we chose variables that could be predicted with a high prediction accuracy (Verbal Recall II, CVLT Short Delay Free Recall, WMS Digit Span). For the latent factor experiments, we used seven behavioral measures (Verbal Recall II, CVLT Long Delay Free Recall, Verbal Recall I, CVLT Short Delay Free Recall, WMS Symbol Span, WMS Digit Span, WAIS Letter-Number Sequencing), which are all correlated with memory.

| HCP Behavior | full | comp | mean | imputePCA | missForest |
|---|---|---|---|---|---|
| PMAT | 0.408 | 0.379 | 0.361 | **0.394** | 0.384 |
| ReadEng | 0.394 | 0.378 | 0.364 | **0.394** | 0.389 |
| PicVocab | 0.457 | 0.432 | 0.413 | **0.433** | 0.430 |
| $1^{st}$pc | 0.565 | NA | 0.512 | **0.519** | **0.519** |

| CNP Behavior | full | comp | mean | imputePCA | missForest |
|---|---|---|---|---|---|
| Verbal Recall II | 0.416 | 0.392 | 0.378 | **0.405** | 0.403 |
| CVLT Short | 0.377 | 0.356 | 0.351 | **0.371** | 0.368 |
| WMS Digit Span | 0.308 | 0.293 | 0.284 | 0.297 | **0.298** |
| $1^{st}$pc | 0.520 | NA | 0.456 | 0.478 | **0.484** |

*Table 1.* rCPM data performance with different data imputation methods for each tested behavioral variable averaged over all missing data percentage. Bolded values indicate the best performing data imputation method for each tested behavioral variable.

methods also had a lower $nrmse$. Results of other behavioral measures were similar to PMAT and Verbal Recall II (see Table 1).

### 3.3. Predicting latent variable

rCPM using **imputePCA** or **missForest** for data imputation significantly improved prediction performance compared to the one using mean imputation as shown in Fig. 2. In HCP dataset, **imputePCA** and **missForest** performed similarly (Fig. 2A); while, **missForest** performed better in CNP dataset (Fig. 2B) at high missing data rates.

## 4. Discussion and Conclusion

We used data imputation to improve prediction performance when behavioral data are missing. Imputation embedded rCPM using either **imputePCA** or **missForest** significantly outperforms simpler methods for handling missing data, such as only using complete cases or mean imputation. In general, ImputePCA for data imputation performed the best.

To the best of our knowledge, this is the first neuroimaging-based predictive modeling study to focus on data imputation for the variable to be predicted (*i.e.*, the behavioral variable). Previous works have only explored data imputation for missing imaging data (Vaden et al., 2012; Thung et al., 2018; Yuan et al., 2012). This framework may have utility for longitudinal studies, where imaging is performed at baseline and behavioral data is collected at multiple future time points. Participant attrition over time is often a major source of missing data in this type of study.

Future work will include using both the imaging and behavioral data to impute missing behavioral data and testing for cases where the data is not missing completely at random. Overall, our results suggest that data imputation may be valuable for CPM studies with missing behavioral data.

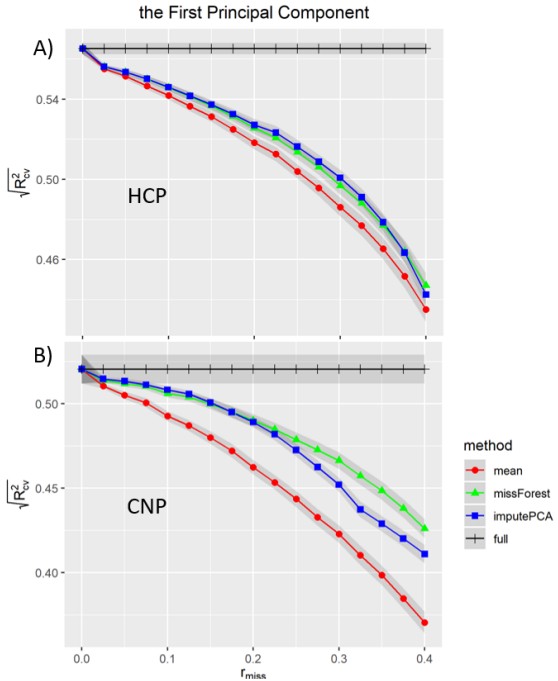

*Figure 2.* Performance rCPM when using data imputation in predicting a latent factor (*i.e.*, the $1^{st}$ principal component) of all behavioral measures in A) HCP dataset and B) CNP dataset over a range of missing data rates from 2.5% to 40% missing data. The shadow areas represent the 95% confidence interval calculated from multiple repeats of missing data.

### 3.2. Predicting single behavioral measure

As a baseline for further comparisons, model performance for predicting single behavioral measure decreased as the percentage of missing data increases. As shown in Fig. 1, rCPM using **imputePCA** or **missForest** for data imputation improved model performance compared to the complete case study. Notably, training model using mean imputation harmed rCPM performance. These two data imputation

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
