# OpenReview forum: "Imputation of Missing Behavioral Measures in Connectome-based Predictive Modelling"
_ICML.cc/2020/Workshop/Artemiss — ICML Artemiss 2020_

### Official Review · AnonReviewer1 · 2020-06-20
**Three data imputation strategies to perform connectome based predictive modelling is presented**

**Rating:** 8
**Confidence:** 4

**Review:**

Summary:
Data imputation strategies to predict missing target variables is studied in this work. Three imputation methods are used to impute missing behavioural data in connectome based predictive modelling. 10 fold CV experiments show promising results.

Strengths:
+ Simple and effective method of imputing missing data for a complex connectome based predictive modelling
+ Experiments are thorough

Concerns:
- Interpretation of the measures used to compare are not explained. For instance, PMAT is not expanded. More importantly a notion of what these measures capture can be useful to present.
- Can the authors motivate why the simulated missing data done randomly? Is there a more structured manner of performing this based on how the data is actually missing.

Typos:
- "we simulated missing " ->
"we simulated the missing data..."

- Title of 3.2 "Predicting"

---

### Decision · Program_Chairs · 2020-07-02

**Decision:**

Accept

**Comment:**

We are very happy to inform you that your paper has been accepted for the Artemiss workshop. We will contact you soon to inform you about the details concerning the format of your presentation at the workshop, and the camera-ready version deadline. Please take into account the referee's comments to write the camera-ready version.